# Analysis of mortality metrics associated with a comprehensive range of disorders in Denmark, 2000 to 2018: A population-based cohort study

Oleguer Plana-Ripoll[1,2]*, Julie W. Dreier[1,3], Natalie C. Momen[1], Anders Prior[4,5], Nanna Weye[1], Preben Bo Mortensen[1,6,7], Carsten B. Pedersen[1,6,7,8], Kim Moesgaard Iburg[5], Maria Klitgaard Christensen[1,5], Thomas Munk Laursen[1], Esben Agerbo[1,6,7], Marianne G. Pedersen[1,6,7], Jørgen Brandt[9,10], Lise Marie Frohn[9], Camilla Geels[9], Jesper H. Christensen[9], John J. McGrath[1,11,12]

1 National Centre for Register-based Research, Aarhus University, Aarhus, Denmark, 2 Department of Clinical Epidemiology, Aarhus University and Aarhus University Hospital, Aarhus, Denmark, 3 Department of Clinical Medicine, University of Bergen, Norway, 4 Research Unit for General Practice, Aarhus, Denmark, 5 Department of Public Health, Aarhus University, Aarhus, Denmark, 6 The Lundbeck Foundation Initiative for Integrative Psychiatric Research (iPSYCH), Aarhus, Denmark, 7 Centre for Integrated Register-based Research at Aarhus University, Aarhus, Denmark, 8 Big Data Centre for Environment and Health, Aarhus University, Aarhus, Denmark, 9 Department of Environmental Science, Aarhus University, Roskilde, Denmark, 10 iClimate, Interdisciplinary Centre of Climate Change, Aarhus University, Roskilde, Denmark, 11 Queensland Centre for Mental Health Research, The Park Centre for Mental Health, Queensland, Australia, 12 Queensland Brain Institute, University of Queensland, St Lucia, Queensland, Australia

* opr@econ.au.dk

## Abstract

### Background

The provision of different types of mortality metrics (e.g., mortality rate ratios [MRRs] and life expectancy) allows the research community to access a more informative set of health metrics. The aim of this study was to provide a panel of mortality metrics associated with a comprehensive range of disorders and to design a web page to visualize all results.

### Methods and findings

In a population-based cohort of all 7,378,598 persons living in Denmark at some point between 2000 and 2018, we identified individuals diagnosed at hospitals with 1,803 specific categories of disorders through the International Classification of Diseases-10th Revision (ICD-10) in the National Patient Register. Information on date and cause of death was obtained from the Registry of Causes of Death. For each of the disorders, a panel of epidemiological and mortality metrics was estimated, including incidence rates, age-of-onset distributions, MRRs, and differences in life expectancy (estimated as life years lost [LYLs]). Additionally, we examined models that adjusted for measures of air pollution to explore potential associations with MRRs. We focus on 39 general medical conditions to simplify the presentation of results, which cover 10 broad categories: circulatory, endocrine, pulmonary, gastrointestinal, urogenital, musculoskeletal, hematologic, mental, and neurologic

**Data Availability Statement:** The individual-level data used for this study are not publicly available, but can be obtained by application to The Danish

Health Data Authority (www.
sundhedsdatastyrelsen.dk). All code and summary
data is available on Open Science Framework
(https://osf.io/zafhu).

**Funding:** This study was supported by the Danish
National Research Foundation, via a Niels Bohr
Professorship to JM. OP-R is supported by a
Lundbeck Foundation Fellowship (R345-2020-
1588) and has received funding from the European
Union's Horizon 2020 research and innovation
programme under the Marie Sklodowska-Curie
grant agreement No 837180. AP is supported by a
grant from the Novo Nordisk Foundation (grant
NNF18OC0031194). The air pollution modelling
was partly funded by NordForsk under the Nordic
Programme on Health and Welfare project #75007
(NordicWelfAir). The Danish Big Data Centre for
Environment and Health is funded by the Novo
Nordisk Foundation Challenge Programme (grant
NNF17OC0027864). The funders had no role in
study design, data collection and analysis, decision
to publish, or preparation of the manuscript.

**Competing interests:** I have read the journal's
policy and the authors of this manuscript have the
following competing interests: JWD has been
involved in projects funded by the Novo Nordisk
Foundation (NNF16OC0019126), The Central
Denmark Region, and the Danish Epilepsy
Association during the conduct of this study.

**Abbreviations:** CI, confidence interval; GATHER,
Guidelines for Accurate and Transparent Health
Estimates Reporting; GBD, Global Burden of
Disease; ICD-10, International Classification of
Diseases-10th Revision; IQR, interquartile range;
LYLs, life years lost; MRR, mortality rate ratio; $NO_2$,
nitrogen dioxide; SMR, standardized mortality
ratio; YLLs, years of life lost.

conditions and cancer. A total of 3,676,694 males and 3,701,904 females were followed up for 101.7 million person-years. During the 19-year follow-up period, 1,034,273 persons (14.0%) died. For 37 of the 39 selected medical conditions, mortality rates were larger and life expectancy shorter compared to the Danish general population. For these 37 disorders, MRRs ranged from 1.09 (95% confidence interval [CI]: 1.09 to 1.10) for vision problems to 7.85 (7.77 to 7.93) for chronic liver disease, while LYLs ranged from 0.31 (0.14 to 0.47) years (approximately 16 weeks) for allergy to 17.05 (16.95 to 17.15) years for chronic liver disease. Adjustment for air pollution had very little impact on the estimates; however, a limitation of the study is the possibility that the association between the different disorders and mortality could be explained by other underlying factors associated with both the disorder and mortality.

## Conclusions

In this study, we show estimates of incidence, age of onset, age of death, and mortality metrics (both MRRs and LYLs) for a comprehensive range of disorders. The interactive data visualization site (https://nbepi.com/atlas) allows more fine-grained analysis of the link between a range of disorders and key mortality estimates.

## Author summary

### Why was this study done?

- There have been many studies related to mortality linked to particular disorders, but these studies have not covered a comprehensive range of disorders.

- Previous studies have traditionally focused on relative measures of mortality (e.g., mortality rate ratios [MRRs]) or crude estimates of life expectancy that do not incorporate variation in the age of onset of the disorder.

- Here, the researchers address these issues in a comprehensive atlas of mortality estimates based on Danish registers.

### What did the researchers do and find?

- Based on 7,378,598 persons living in Denmark in 2000 to 2018, the researchers used national registers to identify individuals diagnosed with 1,803 specific categories of disorders.

- For each of these disorders, a panel of epidemiological and mortality metrics was estimated, including incidence rates, age-of-onset distributions, MRRs, and life years lost (LYLs).

- Within a set of 39 selected medical conditions, mortality rates were larger and life expectancy shorter for 37 conditions compared to the Danish general population.

- The researchers have prepared an interactive data visualization to optimize the interrogation of their findings (http://nbepi.com/atlas).

### What do these findings mean?

- This study allows a more fine-grained analysis of the associations between a comprehensive set of disorders and mortality-related estimates.

- These findings can guide health research and serve as a benchmark to evaluate future health interventions.

## Introduction

Mortality is arguably the most definitive measure of health. Within the field of epidemiology, mortality metrics are of the utmost importance and serve as a foundation for decision-making and prioritization of resources in the healthcare sector. A range of mortality-related metrics are available, such as age-specific mortality rates, standardized mortality ratios (SMRs), mortality rate ratios (MRRs), or case fatality rates. These "relative risk" types of mortality estimates are informative, but should be complemented with measures that examine premature mortality on an absolute scale. In particular, estimates looking at reduction in life expectancy for those experiencing a particular condition tend to be more widely understood by the general community and policy makers. Metrics that link health disorders with premature mortality can inform decision-making on the distribution of limited resources and be used in evaluations of the effectiveness of healthcare provision [1,2].

When looking at premature mortality, the Global Burden of Disease (GBD) studies measure years of life lost (YLLs) [3], which estimates the potential YLLs. The essential feature of this mortality metric is that it is based on the single primary cause of death. However, from a public health perspective, important clues related to prevention may lie many years "upstream" of the cause of death. There is a need to better understand the impact on life expectancy for nonfatal disorders (i.e., disorders that are rarely considered the primary cause of death), as early and more effective treatment of these disorders could reduce premature mortality.

While YLL focuses on age at death, other methods focus on remaining life expectancy at age at diagnosis, which allows us to explore the association between the onset of a broad range of conditions (fatal and nonfatal health outcomes) and subsequent life expectancy. Until recently, most studies linking disorders and life expectancy have applied assumptions related to age of onset of the disorder and/or age of the cohort for follow-up. For example, studies estimating life expectancy have assumed a fixed age of onset of 15 years old for those with mental disorders [4,5], 20 years old for those with type 1 diabetes [6], and 55 years old for those with colon cancer [7]. However, this simplifying assumption can bias estimates of life expectancy. Fortunately, advances in mortality metrics can now take into account the observed age of onset of the disorder of interest [8–10]. To date, this method has been applied mainly to mental disorders [9,11–14], indicating large reductions in life expectancy.

By examining both relative risk–based mortality measures and the more easily interpretable absolute measures of life expectancy, the research community can access a more informative set of mortality-related metrics. Some late-onset disorders among elderly are associated with

large relative mortality risks but small reductions in life expectancy. Conversely, early-onset disorders in younger age groups may be associated with a greater number of life years lost (LYLs) even with modest relative mortality risks. When information on the prevalence and age of disorder onset is also provided, panels that link different types of mortality estimates can provide a richer and more nuanced understanding of the epidemiological landscape describing the association between disorders and mortality.

There is a large literature on mortality-related estimates; however, previous studies have tended to focus on (a) specific prior nonfatal disorders or risk factors and specific causes of death; or (b) a limited range of mortality-related estimates [3]. Furthermore, differences between study designs complicate direct comparison of mortality between different disorders. Thus, there is a need to harmonize mortality measures across a comprehensive range of health disorders and, for each of these disorders, to provide a broader panel of mortality-related estimates, combined with key epidemiological measures, such as incidence or age of onset. The aim of this study was to use the Danish population–based registers to provide a panel of mortality metrics associated with a comprehensive range of health disorders, covering 1,803 different health conditions. In light of a recent Danish study linking exposure to air pollution to increased mortality rates [15], we also undertook a planned sensitivity analysis in order to explore if exposure to air pollution influenced the strength of the association between these disorders and mortality.

## Methods

A protocol was preregistered before having access to the data [16] (S2 Text), and a web page has been designed to visualize all results (http://nbepi.com/atlas).

### Study population and follow-up

We designed a population-based cohort study including all 7,378,598 persons living in Denmark at any point between January 1, 2000 and December 31, 2018 (all individuals were included regardless of whether they had a hospital contact or not). Since 1968, the Danish Civil Registration System [17] has maintained information on all residents, including sex, date of birth, continuously updated information on vital status, and a unique personal identification number that can be used to link information from national registers.

### Assessment of specific disorders based on ICD-10 classification

Specific disorders were identified through hospital contacts on or after January 1, 1995, allowing a period of at least 5 years to identify individuals with diseases diagnosed before the start of the follow-up. This information was obtained from the Danish National Patient Register [18], which contains data on all admissions to hospital inpatient facilities and visits to outpatient facilities (including visits to medical specialists), as well as emergency departments, since 1995 (including contacts from psychiatric departments, available through the Danish Psychiatric Central Research Register) [19]. The diagnostic system used during this period was the Danish modification of the International Classification of Diseases-10th Revision (ICD-10) [20]. For this study, we considered 19 overall chapters (e.g., A00-B99: Certain infectious and parasitic diseases), 207 subchapters (e.g., A00-A09: Intestinal diseases), and 1,538 3-character categories for certain disorders (e.g., A00: Cholera), making a total list of 1,764 specific categories. Additionally, we considered 39 general medical conditions (10 broad categories, 8 of which included 29 subcategories) based on combinations of several ICD-10 codes previously used in Danish health research [21,22]. Thus, the total list of disorders and related ICD-10 codes for which mortality-related metrics were estimated comprised 1,803 categories and is available in

the Supporting information (S1 Table). For each individual in the study, the date of onset for each disorder was defined as the date of first contact (inpatient, outpatient, or emergency visit) for the specific disorder (different disorders developing within the same individual could have different dates of onset).

## Mortality

Information on date and primary cause of death was obtained from the Danish Registry of Causes of Death [23]. All deaths were categorized in 2 widely used and nonoverlapping groups according to ICD-10 codes: external causes of death, which included suicide (X60-X84 and Y87.0), homicide (X85-Y09 and Y87.1), and accidents (V01-X59, Y10-Y86, Y87.2, and Y88-Y89), and natural causes of death, which included all other causes.

## Statistical analysis

All individuals were followed up from birth, immigration to Denmark, or January 1, 2000, whichever came last, until death, emigration from Denmark, or December 31, 2018, whichever came first. All disorders were treated as time-varying factors (additional details in S1 Text). A list of all epidemiological measures reported in this study for each of the 1,803 disorders is available in Box 1. A description of the methods to estimate mortality rates and life expectancy

### Box 1. Epidemiological measures reported for each of the 1,803 health disorders

**Number of diagnosed**: Number of individuals living in Denmark at some point between 2000 and 2018 diagnosed with the specific disorder at a hospital between January 1, 1995 and December 31, 2018.

**Age at diagnosis**: Median and IQR of age at diagnosis among those diagnosed with the specific disorder.

**Number of deaths**: Number of individuals who died between January 1, 2000 and December 31, 2018 after having received a diagnosis of the disorder.

**Age at death**: Median and IQR of age at death among those diagnosed with the specific disorder who died during the study period.

**Incidence rates**: Number of individuals diagnosed with the disorder for the first time per unit of time. In this study, incidence rates are reported per 10,000 person-years for each age group (0 to 5, 5 to 10, 10 to 15, . . ., and 95 to 100 years).

**Mortality rates**: Number of deaths per unit of time. In this study, mortality rates are reported per 10,000 person-years for those diagnosed with a specific disorder and for the entire population standardized to the same sex and age of those diagnosed with the disorder. Results are shown for all ages and for each age group (0 to 5, 5 to 10, 10 to 15, . . ., and 95 to 100 years).

**MRRs**: Represent the ratio of mortality rates between persons with and without a diagnosis of the specific disorder adjusted for age, sex, and birth date. In this study, we report MRRs for (i) all causes of death and separately for natural and external causes; (ii) males and females combined and separately; and (iii) overall MRRs as well as MRRs depending

on age and time since the first diagnosis. Models including air pollution adjusted also for mean $NO_2$ and $PM_{2.5}$ during the year before start of follow-up.

**Average life expectancy**: Represents the average number of years a person is expected to live if age-specific mortality rates in a given period remain constant in the future. In this study, we report average remaining life expectancy at specific ages for individuals previously diagnosed with a specific disorder and for the general population.

**LYLs after diagnosis**: Represent differences in remaining life expectancy between individuals diagnosed with a specific disorder and the general population of same age and sex. In this study, we report LYLs for (i) all causes of death and separately for natural and external causes; and (ii) males and females combined and separately.

is provided below, while specific details for all other measures are available in the Supporting information (S1 Text). All metrics were estimated for males and females separately and combined. Individuals with more than one diagnosis contributed information for each of their diagnoses; however, only information in relation to the specific disorder was considered in the estimates.

Mortality rates for the whole population and for those diagnosed with each disorder were calculated as the number of deaths divided by the total follow-up time in person-years. Standardized mortality rates for the whole population were calculated using the distribution of sex, age (5-year categories), and calendar time (2000 to 2004, 2005 to 2009, 2010 to 2014, and 2015 to 2018) of those diagnosed with each disorder. MRRs with 95% confidence intervals (CIs) were estimated for external and natural causes of death and for all-causes combined, comparing persons with and without each specific disorder using Cox proportional hazards models, with age as the underlying timescale, and adjusting for sex and birth date (using cubic splines with 4 knots). MRRs for all causes depending on age (5-year categories) and time since diagnosis (0 to 6 months, 6 to 12 months, 1 to 2 years, 2 to 5 years, 5 to 10 years, and 10+ years) were estimated including an interaction term with exposure in the regression models.

Differences in average life expectancy between the group of persons with a specific disorder and the general population were calculated as LYLs. The technical development of this method has recently been published [8,9], and a detailed account of how to implement it—with a specific R package—is available [10]. In brief, for each disorder, the expected residual lifetime was calculated at each possible age of diagnosis for the group of persons with a previous diagnosis and for the general population of same sex and age based on age-specific mortality rates. The main reason to compare those with a given disease to the general population—and not to persons without the disease—is that the number of LYLs at a given age, e.g., 45 years, is estimated using mortality rates at ages 45 years and beyond. By choosing persons without the disease as a comparison group, we would assume that someone who has not experienced the disease at age 45 would remain free of the disease until death. Although it might seem problematic to include persons with a disease in both the diseased and reference groups, this is analogous to SMRs, which compare mortality in a group of persons to the one in the general population. The difference between the estimate for those with a diagnosis and the general population was defined as differences in life expectancy at each possible age of diagnosis, and it requires the assumption that those diagnosed will experience the mortality rates of the diagnosed during the entire life (after diagnosis). A weighted average of all these age-specific estimates (weighted by the number of individuals diagnosed at each age) provided a summary measure of differences in

life expectancy after disorder diagnosis. Finally, these differences were divided into natural and external causes of death using a competing risks model [24]. CIs for these estimates were obtained using nonparametric bootstrap with 500 iterations.

Our comprehensive and multifaceted approach provided us with the opportunity to explore the influence of candidate risk factors on mortality-related estimates. As air pollution is a prominent environmental health threat, we examined the potential confounding effect of air pollution on MRRs. The study population was linked with information on residential exposure to levels of nitrogen dioxide ($NO_2$) and atmospheric particulate matter with a diameter of less than 2.5 micrometers ($PM_{2.5}$) modeled using the multiscale and integrated air pollution model system [25,26] during the year before start of follow-up (specific details available in S1 Text). Models estimating MRRs for all-cause mortality were replicated with and without adjustment for mean $NO_2$ and $PM_{2.5}$, included in the models as continuous z-scores. The models did not adjust for socioeconomic characteristics, as data at the individual level could not be used.

## Preregistered protocol, code and data availability, and visualization of results

This study is reported as per the Guidelines for Accurate and Transparent Health Estimates Reporting (GATHER) guideline (checklist available in S2 Table). The protocol and analysis plan (S2 Text) were posted on Open Science Framework before having access to the data and are publicly available together with all programming code and summary outcomes [16]. Due to data protection laws, researchers have to apply to the Danish Health Data Authority to have access to the underlying person-level data.

We have developed an interactive web page to visualize all results from this study [27]. In order to simplify the presentation of results, we focus on the 39 general medical conditions [21,22], which cover 10 broad categories: circulatory, endocrine, pulmonary, gastrointestinal, urogenital, musculoskeletal, hematologic, mental, and neurologic conditions and cancer (see S3 Table for details). Individuals were considered to experience one of the broad categories if they were diagnosed with at least one of the disorders included in the category. Complete results for each of the 1,803 specific disorders are available on Open Science Framework [16] and on the web page (see S1 Fig for an overview of the main results for one specific disorder).

All analyses were performed on the secured platform of the Danish Health Data Authority. The Danish Data Protection Agency and the Danish Health Data Authority approved this study. According to Danish law, informed consent is not required for register-based studies. All data accessed were deidentified.

## Results

A total of 7,378,598 persons (3,676,694 males and 3,701,904 females) were included in the study and followed up for 101.7 million person-years. The mean (standard deviation) age at entry to and exit from the study was 31.2 (24.4) and 44.9 (25.7) years, respectively. During the 19-year follow-up period, 1,034,273 persons (14.0%) died (509,032 males [13.8%] and 525,241 females [14.2%]), and 989,770 did so of natural causes and 44,503 of external causes. When focusing on the 39 selected disorders, the most prevalent were disorders of the circulatory system ($n = 1,431,041$), disorders of the neurological system ($n = 1,283,880$), and mental disorders ($n = 1,128,977$) (Fig 1). The median (interquartile range, IQR) age at diagnosis ranged from 34.0 (18.1 to 50.7) years for allergy to 76.2 (66.9 to 83.7) years for heart failure, whereas median (IQR) age at death for those diagnosed ranged from 53.5 (44.8 to 63.9) years for HIV/AIDS to 86.3 (79.7 to 91.4) years for hearing problems (Fig 1).

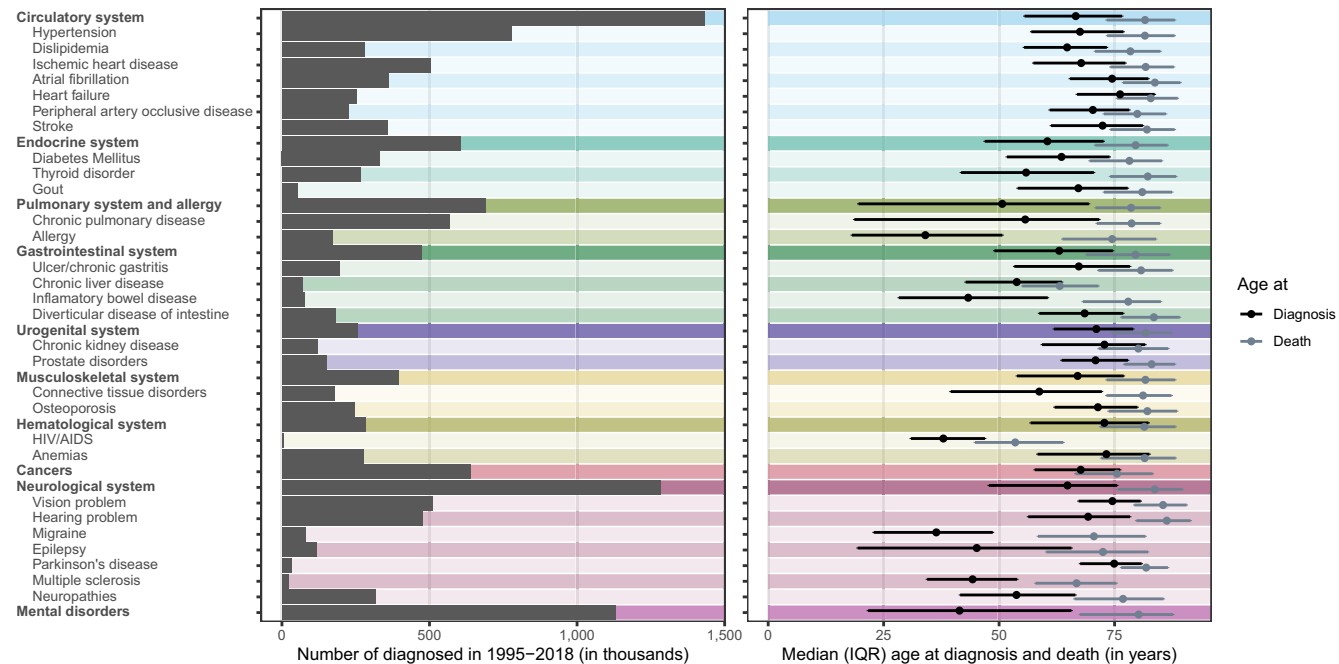

**Fig 1.** Number of diagnosed (left panel) and age at diagnosis and death among the diagnosed (right panel) for 39 selected conditions covering 10 broad categories. Estimates are available in S4 Table. IQR, interquartile range.

## All-cause mortality

For 37 of the 39 disorders, mortality rates were larger and life expectancy shorter compared to the Danish general population (Fig 2). For these 37 disorders, MRRs ranged from 1.09 (1.09 to 1.10) for vision problems to 7.85 (7.77 to 7.93) for chronic liver disease, while reduction of life expectancy ranged from 0.31 (0.14 to 0.47) years (approximately 16 weeks) for allergy to 17.05 (16.95 to 17.15) years for chronic liver disease. The remaining 2 disorders were hearing problems, with reduced mortality rates (MRR = 0.93 [0.93 to 0.94]) and slightly longer life expectancy (LYL = −0.11 [−0.15 to −0.07] years; approximately 40 days); and migraine, with similar mortality rates (MRR = 1.00 [0.97 to 1.03]) and slightly longer life expectancy (LYL = −0.65 [−0.94 to −0.37] years). The adjustment for air pollution had very little impact on the estimates (i.e., the sign and the magnitude of the effect size did not vary, and the CIs between adjusted and unadjusted estimates were comparable; S2 Fig). The combination of different metrics is useful to show differences between disorders. For example, we observed similar increases in mortality rates in those with a disorder of the circulatory system (MRR = 2.90 [2.89 to 2.92]) and those with chronic pulmonary disease (MRR = 2.94 [2.92 to 2.95]), but the reduction in life expectancy was 8.27 (8.21 to 8.32) years in those diagnosed with the latter and 3.80 (3.78 to 3.82) years in those diagnosed with the former.

## Cause-specific and sex-specific mortality

Estimates depending on natural and external causes of death are available in S3 Fig. MRRs due to natural causes of death were larger than those for external causes in 25 of the 39 disorders. For all disorders except mental disorders (LYL = 2.05 [2.00 to 2.10] years), chronic liver disease (LYL = 1.87 [1.75 to 1.99] years), and epilepsy (LYL = 1.21 [1.10 to 1.33] years), the reduction

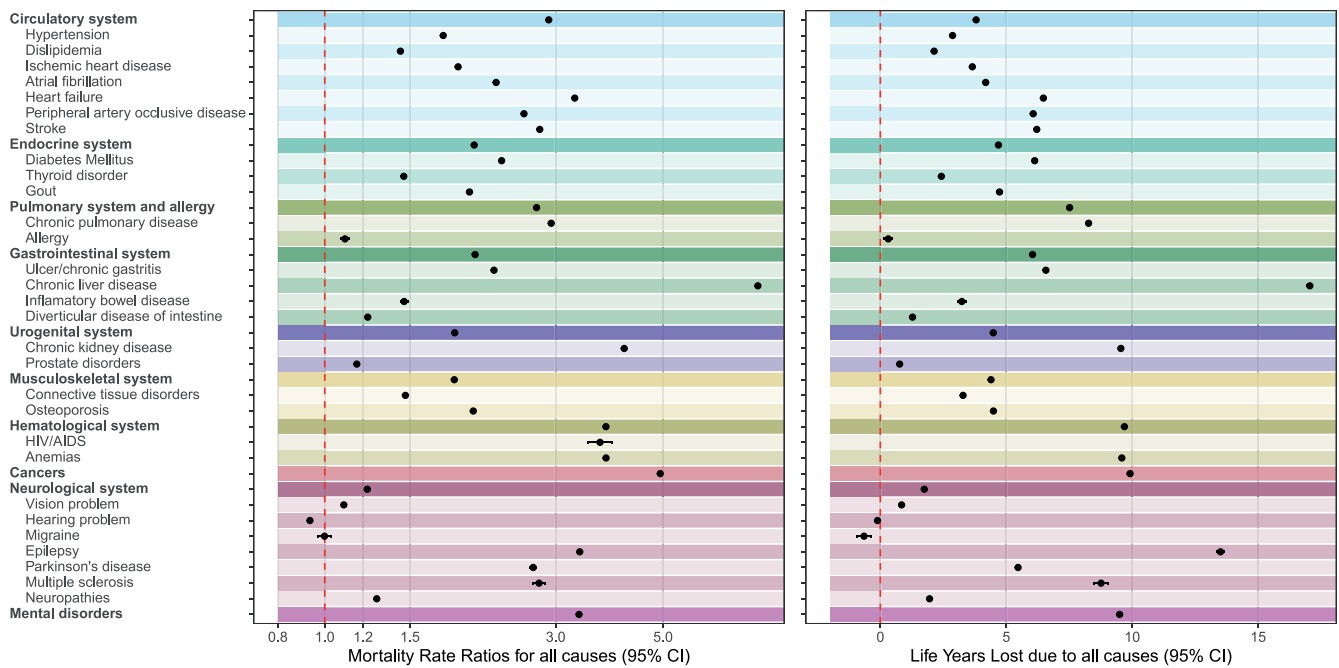

**Fig 2.** MRRs (left panel) and LYLs (right panel) for all causes of death for 39 selected conditions covering 10 broad categories. The red line indicates equal mortality in the 2 groups (MRR of 1; LYLs of 0). Estimates are not shown if they are based on less than 100 individuals diagnosed or less than 20 deaths; for LYLs, estimates are not shown if there were not enough individuals at older ages of follow-up. Estimates are available in S4 Table. CI, confidence interval; LYLs, life years lost; MRR, mortality rate ratio.

in life expectancy explained by external causes of death was less than 1 year compared to the general population (LYLs related to external causes in these disorders ranged from −0.41 [−0.42 to −0.40] years for cancers to 0.62 [0.55 to 0.69] years for chronic kidney disease). For most disorders, MRRs and LYLs were larger in males than in females or similar for both sexes (sex-specific estimates are available in S4 Fig).

## MRRs depending on age and time since onset of the disorder

Age- and time-specific MRRs are available for the 10 broad categories of disorders in Figs 3 and 4. When looking at age-specific MRRs (Fig 3), generally, the largest MRRs were found in children and young adults. However, the decline in MRRs was not always linear with age, with some disorders (e.g., gastrointestinal or mental disorders) showing modest increases before declines in older adults. For most disorders, MRRs were largest in the first 6 months after diagnosis (Fig 4). For some disorders (e.g., cancer or musculoskeletal), there was a continuous decrease after diagnosis, while for some other disorders (e.g., circulatory or mental disorders), there was a decrease, which remained constant after the first 6 months post-diagnosis. Mortality rates 10 years after diagnosis remained higher for those with a diagnosis of any of the disorders, compared to those without the diagnosis: MRRs 10 years after diagnosis ranged from 1.27 (1.26 to 1.28) for disorders of the neurological system to 3.18 (3.16 to 3.21) for mental disorders.

## Estimates for all ICD-10 disorders

In addition to the 39 disorders discussed above, we have provided estimates for all 1,803 disorders on Open Science Framework [16] and on the interactive web page [27]. For example, Fig 5 shows the number of diagnosed, MRRs and LYLs for all 1,538 ICD-10 3-character categories

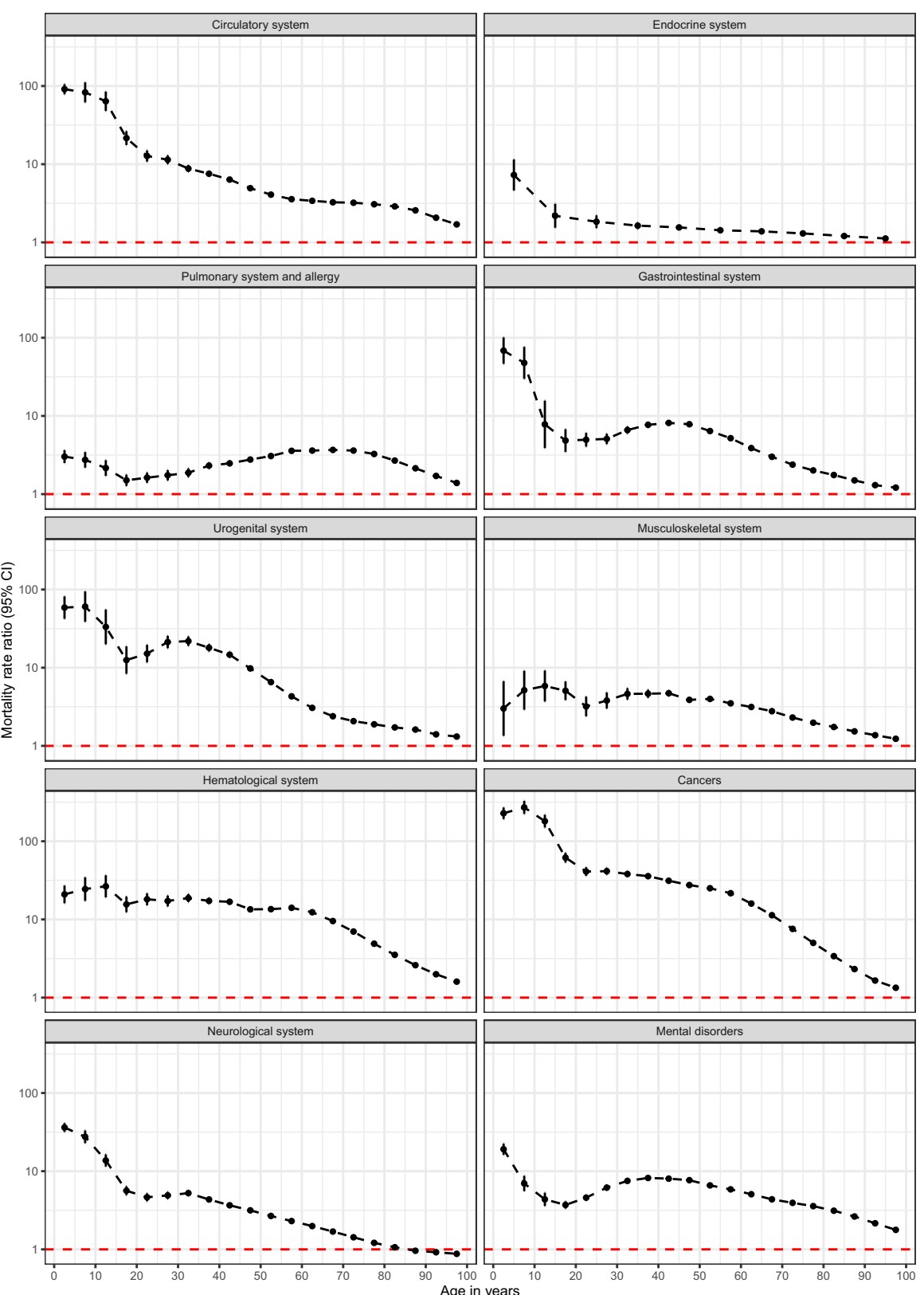

**Fig 3. Age-specific mortality rates ratios adjusting for age, sex, and birth date for 10 broad categories of conditions.** The red line indicates equal mortality in the 2 groups (MRR of 1). Estimates are available on Open Science Framework [16] and on the interactive web page [27]. CI, confidence interval; MRR, mortality rate ratio.

of disorders as well as the overall results for each of the 19 chapters. When looking at overall ICD-10 chapters (Fig 5A), MRRs ranged from 0.63 (0.61 to 0.65) for Chapter XV (Pregnancy, childbirth and the puerperium) to 4.14 (4.12 to 4.15) for Chapter X (Diseases of the respiratory system), while LYLs ranged from 0.29 (0.24 to 0.35) years for Chapter VIII (Diseases of the ear and mastoid process) to 13.90 (12.97 to 14.93) years for Chapter XVI (Certain conditions originating in the perinatal period). When looking within ICD chapters (Fig 5B), there is considerable heterogeneity, with MRRs ranging from 0.50 (0.38 to 0.64) for ICD-10 code O83 (Other assisted single delivery; Chapter XV) to 152.31 (116.61 to 198.93) for P60 (Disseminated intravascular coagulation of newborn; Chapter XVI) and LYLs ranging from −3.37 (−3.48 to −3.24) years for M23 (Internal derangement of knee; Chapter XIII) to 26.95 (25.43 to 28.58) years for C74 (Malignant neoplasm of adrenal gland; Chapter II).

## Discussion

Our study provides a comprehensive atlas of mortality-related estimates, based on high-quality Danish registers. To the best of our knowledge, this is the most detailed compendium of mortality-related estimates and the first to show reductions in life expectancy for a comprehensive range of disorders. The interactive data visualization site allows more fine-grained analysis of the association between a range of disorders and key mortality-related estimates. The discussion will focus on 3 key points, and when necessary, we will illustrate these points using the 39 conditions described above.

First, our findings are broadly consistent with related publications [3,28]. The majority of disorders were associated with both an increased mortality rate and a reduction in life expectancy of at least 1 year. However, important nuances were revealed when the 2 main mortality-related estimates are examined together. For example, we observed similar increases in mortality rates in those with a disorder of the circulatory system and those with chronic pulmonary disease; however, the reduction in life expectancy was double in those diagnosed with the latter compared to those diagnosed with the former. The discrepancy in those pairs of estimates is most likely explained by differences in age of onset. The median age of register-based onset for those diagnosed with chronic pulmonary disease was 55.7 years; thus, individuals with this disorder have more years of potential life lost compared to those diagnosed with a disorder of the circulatory system, with a median age of onset of 66.6 years.

Second, there were clear temporal signatures between the majority of disorders and time-dependent MRRs—the difference in mortality rates between those with and without a specific disorder peaks in the first 6 months after the diagnosis of the disorder, and it then decreases. For some disorders, there was a sharp decline in MRRs after the first 6 months, and then they were stable for the following 10 years. This may reflect a subgroup of individuals who delay seeking help until the disorder is more advanced. For other disorders, there was a stable decrease over time, suggesting that the disorder may remit over time (spontaneously or in response to optimal treatment). Finally, MRRs did not increase over time after onset for any of the selected 39 conditions; however, MRRs increased over time for few disorders (e.g., dementia; ICD-10 code F00) when looking at all 1,803 disorders. In summary, time-dependent MRRs provide interesting features of the association between specific disorders and mortality risks.

Third, it is important to note that a range of factors can influence mortality in those with specific health conditions. We do not propose a causal relationship between the health

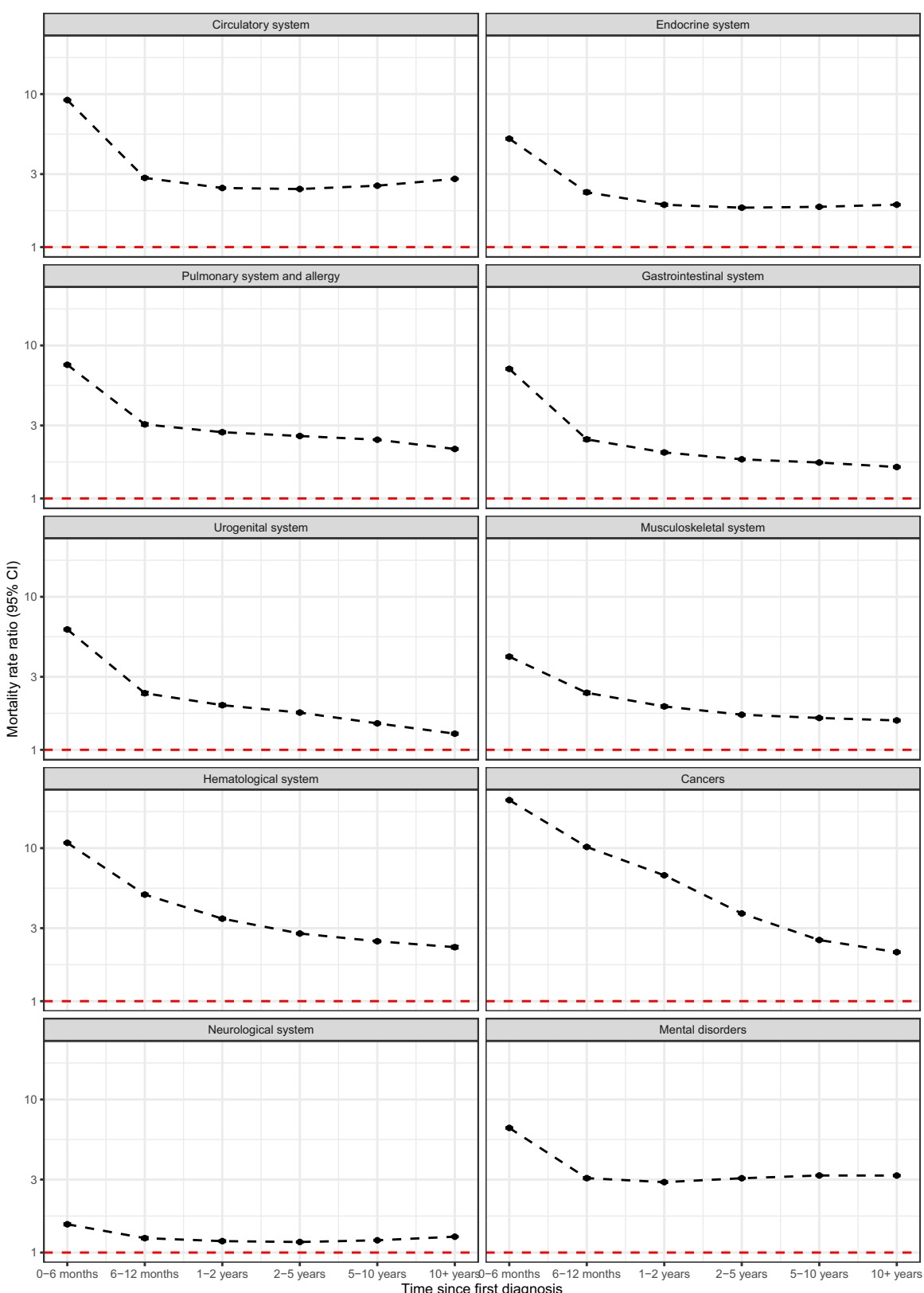

**Fig 4. Mortality rates ratios depending on time since first diagnosis for 10 broad categories of conditions.** Estimates are adjusted for age, sex, and birth date. The red line indicates equal mortality in the 2 groups (MRR of 1). Estimates are available on Open Science Framework [16] and on the interactive web page [27]. CI, confidence interval; MRR, mortality rate ratio.

conditions and subsequent mortality. The observed associations could be explained by underlying factors that are associated both with morbidity and mortality (e.g., socioeconomic or environmental factors). Additionally, the onset of a health condition can have impacts on lifestyle, daily habits, and socioeconomic characteristics, which, in turn, might mediate the association with subsequent mortality. Although air pollution has been found to be associated both with mortality [15,29] and a wide range of health conditions [30,31], we found that the relationship between the disorders and mortality rates was not substantially altered in models adjusting for air pollution exposure. However, air pollutants were modeled at the residential address during only 1 year, and they might be poor indicators of the real individual accumulated exposure (which might occur at the work or educational place, for example). We plan to explore this issue in more detail in future studies.

This study has several strengths. First and foremost, all analyses are based on the same population, same time period, and same methodology, which allows comparison of results between different conditions. Additionally, we used an innovative metric of life expectancy, which can be more readily understood by the general community and that has several advantages. The "Life Years Lost" method uses the age of diagnosis and overcomes limitations from previous studies in which a fixed age of onset had to be assumed [4–7]. Additionally, it allows for the evaluation of the impact of particular disorders on premature mortality, regardless of how the individual died. A recent study using this method based on Danish registers [13] has shown that all mental and substance use disorders are associated with a reduction in life expectancy, although only a few of them are generally included as cause of death. Finally, it allows the total average reduction in life expectancy to be partitioned into specific types of causes of death; while we only used 2 broad categories (natural and external causes), the findings could be divided into narrower cause of death categories specifically selected for each health condition. We plan to explore these additional options in future studies.

While the study is based on a large sample including the entire Danish population and is based on complete Danish registers, it has several limitations. We used age at first diagnosis in the registers as a proxy for age of onset, which could introduce biases for disorders with delayed help-seeking. Individuals diagnosed with a specific disorder include only those diagnosed in hospitals, outpatient, and accident/emergency settings. The study does not include patients with disorders that more likely were treated by the general practitioner or who were not treated at all. While this limitation might have little impact on disorders such as cancer, psychotic disorders, or renal failure, the true prevalence and incidence of disorders like allergy, mild depression, or alcohol dependence might be underestimated, and their associated excess mortality overestimated since the mortality metrics are based on the subset of individuals with more severe diseases that are seen and treated in secondary care. In addition, this study did not include information on remission or other comorbid disorders; the group of individuals with a specific disorder can therefore be interpreted as persons who have had a diagnosis of the disorder regardless of whether they have other disorders or whether they have recovered afterward. Estimates of life expectancy are based on mortality rates from onset and onward; thus, they can be interpreted mostly for chronic conditions. Finally, information on mortality was also obtained from registers. Since date of death is considered to be accurate, all-cause mortality is not affected by potential misclassification. However, there could be some misclassification of the specific cause of death, given that only 5% of deaths in Denmark are examined

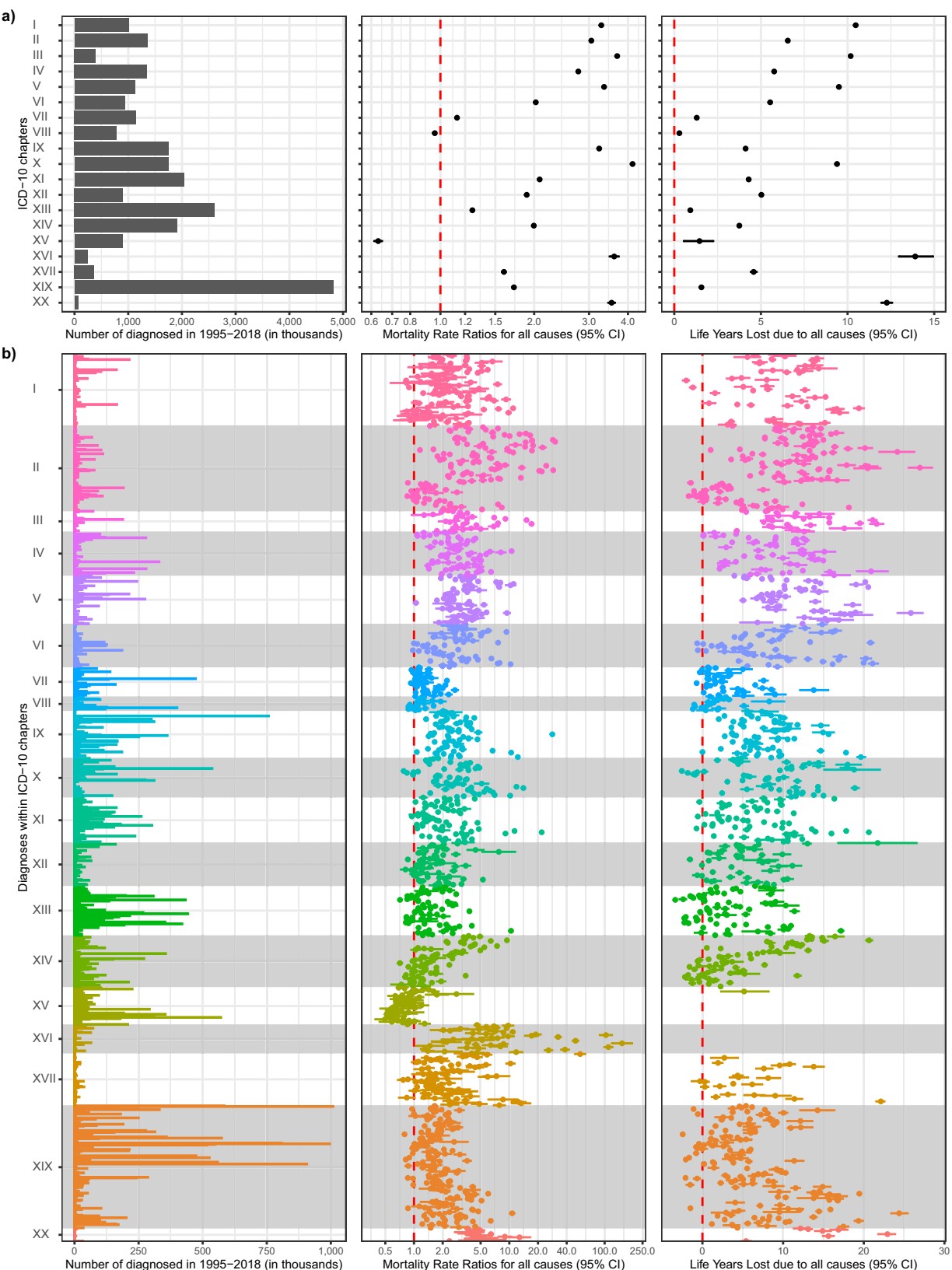

**Fig 5.** Number of diagnosed; MRRs; and LYLs for (a) 19 overall ICD-10 chapters, and (b) 1,538 3-character categories within each chapter. The red line indicates equal mortality in the 2 groups (MRR of 1; LYLs of 0). Estimates are not shown if they are based on less than 100 individuals diagnosed or less than 20 deaths; for LYLs, estimates are not shown if there were not enough individuals at older ages of follow-up. Estimates are available on Open Science Framework [16], but we encourage the reader to explore the estimates in more detail at the interactive web page [27]. The x-axes are different in figures (a) and (b). **Chapters**: (I) Certain infections and parasitic diseases; (II) Neoplasms; (III) Diseases of the blood and blood-forming organs and certain disorders involving the immune mechanism; (IV) Endocrine, nutritional and metabolic diseases; (V) Mental, behavioral and neurodevelopmental disorders; (VI) Diseases of the nervous system; (VII) Diseases of the eye and adnexa; (VIII) Diseases of the ear and mastoid process; (IX) Diseases of the circulatory system; (X) Diseases of the respiratory system; (XI) Diseases of the digestive system; (XII) Diseases of the skin and subcutaneous tissue; (XIII) Diseases of the musculoskeletal system and connective tissue; (XIV) Diseases of the genitourinary system; (XV) Pregnancy, childbirth and the puerperium; (XVI) Certain conditions originating in the perinatal period; (XVII) Congenital malformations, deformations and chromosomal abnormalities; (XIX) Injury, poisoning and certain other consequences of external causes; (XX) External causes of morbidity. CI, confidence interval; ICD-10, International Classification of Diseases-10th Revision; LYLs, life years lost; MRR, mortality rate ratio.

by autopsy [32]. However, given that all deaths were classified into 2 broad categories (natural and external causes), misclassification is less likely. Finally, while our estimates may be reflective of high-income countries, it remains unknown the extent to which our findings generalize to other countries with different healthcare systems or levels of air pollution.

The analytic framework we presented can be used for a range of important public health research issues. For example, with the aging population, we can expect that some late-onset disorders (e.g., dementia) will increase over time. Additionally, the observed mortality estimates can be used for burden of disease modeling exercises.

In conclusion, we have presented a detailed atlas of disease mortality based on Danish hospital registers. Our study has provided a large amount of data, and we would like to emphasize the need for web-based data visualizations tools to make these data available to the broader public. With such interactive websites, it is possible for the reader to inspect specific conditions and have access to summary information of interest (e.g., age of onset, age of death, prevalence, incidence, mortality rates, life expectancy, etc.). We hope that this comprehensive study (with all summary data made available through an interactive data visualization site) can be used to generate future hypothesis-driven research.

## Supporting information

**S1 Text. Supplementary methods.**
(PDF)

**S2 Text. Prespecified analysis plan.**
(PDF)

**S1 Table. List of 1,803 disorders and related ICD-10 codes.** ICD-10, International Classification of Diseases-10th Revision.
(XLSX)

**S2 Table. GATHER checklist.** GATHER, Guidelines for Accurate and Transparent Health Estimates Reporting.
(PDF)

**S3 Table. List of 39 selected conditions covering 10 broad categories of disorders.**
(PDF)

**S4 Table. For 39 selected conditions covering 10 broad categories: number of diagnosed, age at diagnosis, number of deaths among the diagnosed, age at death, MRRs and LYLs for all causes of death.** LYLs, life years lost; MRR, mortality rate ratio.
(PDF)

**S5 Table. MRRs for all causes of death for 39 selected conditions covering 10 broad categories with and without adjustment for air pollution during the year before start of follow-up.** All estimates are adjusted for age, sex, and birth date. MRR, mortality rate ratio.
(PDF)

**S6 Table. MRRs and LYLs for natural and external causes of death for 39 selected conditions covering 10 broad categories.** Estimates are not shown if they are based on less than 100 individuals diagnosed or less than 20 deaths; for LYLs, estimates are not shown if there were not enough individuals at older ages of follow-up. LYLs, life years lost; MRR, mortality rate ratio.
(PDF)

**S7 Table. Number of females and males diagnosed and sex-specific MRRs and LYLs for all causes of death for 39 selected conditions covering 10 broad categories.** Estimates are not shown if they are based on less than 100 individuals diagnosed or less than 20 deaths; for LYLs, estimates are not shown if there were not enough individuals at older ages of follow-up. LYLs, life years lost; MRR, mortality rate ratio.
(PDF)

**S1 Fig. Overview of the main results for mental disorders (ICD-10 F00-F99) available on the website http://nbepi.com/atlas [27].** ICD-10, International Classification of Diseases-10th Revision.
(PDF)

**S2 Fig. MRRs for all causes of death for 39 selected conditions covering 10 broad categories with and without adjustment for air pollution during the year before start of follow-up.** The red line indicates equal mortality in the 2 groups (MRR of 1). All estimates are adjusted for age, sex, and birth date. Estimates are available in S5 Table and on Open Science Framework [16]. MRR, mortality rate ratio.
(PDF)

**S3 Fig. MRRs and LYLs for natural and external causes of death for 39 selected conditions covering 10 broad categories.** The red line indicates equal mortality in the 2 groups (MRR of 1; LYLs of 0). Estimates are not shown if they are based on less than 100 individuals diagnosed or less than 20 deaths; for LYLs, estimates are not shown if there were not enough individuals at older ages of follow-up. Estimates are available in S6 Table and on Open Science Framework [16]. LYLs, life years lost; MRR, mortality rate ratio.
(PDF)

**S4 Fig. Number of females and males diagnosed and sex-specific MRRs and LYLs for all causes of death for 39 selected conditions covering 10 broad categories.** The red line indicates equal mortality in the 2 groups (MRR of 1; LYLs of 0). Estimates are not shown if they are based on less than 100 individuals diagnosed or less than 20 deaths; for LYLs, estimates are not shown if there were not enough individuals at older ages of follow-up. Estimates are available in S7 Table and on Open Science Framework [16]. LYLs, life years lost; MRR, mortality rate ratio.
(PDF)

## Acknowledgments

We would like to thank Sussie Antonsen for data management in relation to air pollution.

## Author Contributions

**Conceptualization:** Oleguer Plana-Ripoll, Julie W. Dreier, Anders Prior, Preben Bo Mortensen, Carsten B. Pedersen, John J. McGrath.

**Data curation:** Oleguer Plana-Ripoll, Julie W. Dreier, Natalie C. Momen, Carsten B. Pedersen, Marianne G. Pedersen, Jørgen Brandt, Lise Marie Frohn, Camilla Geels, Jesper H. Christensen.

**Formal analysis:** Oleguer Plana-Ripoll.

**Funding acquisition:** Oleguer Plana-Ripoll, Preben Bo Mortensen, Esben Agerbo, John J. McGrath.

**Investigation:** Oleguer Plana-Ripoll, Julie W. Dreier, Preben Bo Mortensen.

**Methodology:** Oleguer Plana-Ripoll, Julie W. Dreier, Jørgen Brandt, Jesper H. Christensen.

**Project administration:** Carsten B. Pedersen.

**Resources:** Preben Bo Mortensen, Carsten B. Pedersen, Esben Agerbo.

**Software:** Oleguer Plana-Ripoll.

**Supervision:** Preben Bo Mortensen, Carsten B. Pedersen, John J. McGrath.

**Validation:** Oleguer Plana-Ripoll, Jørgen Brandt, Lise Marie Frohn, Camilla Geels, Jesper H. Christensen.

**Visualization:** Oleguer Plana-Ripoll, Nanna Weye, John J. McGrath.

**Writing – original draft:** Oleguer Plana-Ripoll, Julie W. Dreier, John J. McGrath.

**Writing – review & editing:** Oleguer Plana-Ripoll, Julie W. Dreier, Natalie C. Momen, Anders Prior, Nanna Weye, Preben Bo Mortensen, Carsten B. Pedersen, Kim Moesgaard Iburg, Maria Klitgaard Christensen, Thomas Munk Laursen, Esben Agerbo, Marianne G. Pedersen, Jørgen Brandt, Lise Marie Frohn, Camilla Geels, Jesper H. Christensen, John J. McGrath.

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
