## [Editor Report · Decision Letter 0]

16 Dec 2021

Dear Dr Plana-Ripoll, 

Thank you for submitting your manuscript entitled "An analysis of mortality metrics associated with a comprehensive range of disorders: the Danish atlas of disease mortality" for consideration by PLOS Medicine.

Your manuscript has now been evaluated by the PLOS Medicine editorial staff and I am writing to let you know that we would like to send your submission out for external peer review.

Please re-submit your manuscript within two working days, i.e. by Dec 20 2021 11:59PM.

Kind regards,

Caitlin Moyer, Ph.D.

Associate Editor

PLOS Medicine

---

## [Decision Letter · Decision Letter 1]

10 Mar 2022

Dear Dr. Plana-Ripoll,

Thank you very much for submitting your manuscript "An analysis of mortality metrics associated with a comprehensive range of disorders: the Danish atlas of disease mortality" (PMEDICINE-D-21-05097R1) for consideration at PLOS Medicine. 

Your paper was evaluated by a senior editor and discussed among all the editors here. It was also sent to three independent reviewers, including a statistical reviewer. The reviews are appended at the bottom of this email and any accompanying reviewer attachments can be seen via the link below:

[LINK]

In light of these reviews, I am afraid that we will not be able to accept the manuscript for publication in the journal in its current form, but we would like to consider a revised version that addresses the reviewers' and editors' comments. Obviously we cannot make any decision about publication until we have seen the revised manuscript and your response, and we plan to seek re-review by one or more of the reviewers. 

We expect to receive your revised manuscript by Mar 31 2022 11:59PM. Please email us (plosmedicine@plos.org) if you have any questions or concerns.

We look forward to receiving your revised manuscript. 

Sincerely,

Caitlin Moyer, Ph.D.

Associate Editor

PLOS Medicine

plosmedicine.org

1. Title: Please revise your title according to PLOS Medicine's style. Your title must be nondeclarative and not a question. It should begin with main concept if possible. "Effect of" should be used only if causality can be inferred, i.e., for an RCT. Please place the study design ("A randomized controlled trial," "A retrospective study," "A modelling study," etc.) in the subtitle (ie, after a colon).

2. Line numbers: Please include line numbers running continuously throughout the manuscript with the revised version.

3. Abstract: Methods and Findings: Please include a few examples or broad summary of some of the 39 general medical conditions discussed. For example, as described in the Methods: “...ten broad categories: circulatory, endocrine, pulmonary,gastrointestinal, urogenital, musculoskeletal, hematologic, mental, and neurologic conditions and cancer…”

4. Abstract: Methods and Findings: Please mention which conditions correspond with these MRRs/LYLs given as examples: “For these 37 disorders, MRRs ranged from 1.09 (95%CI: 1.09-1.10) to 7.85 (7.77-7.93), while LYLs ranged from 0.31 (0.14-0.47) years (~16 weeks) to 17.05 (16.95-17.15) years.”

5. Abstract: Methods and Findings: In the last sentence of the Abstract Methods and Findings section, please describe the main limitation(s) of the study's methodology.

6. Abstract: Conclusions: Please address the study implications without overreaching what can be concluded from the data; the phrase "In this study, we observed ..." may be useful. Please avoid or temper assertions of primacy ("To the best of our knowledge, this atlas is the first....").

7. Author summary: At this stage, we ask that you include a short, non-technical Author Summary of your research to make findings accessible to a wide audience that includes both scientists and non-scientists. The Author Summary should immediately follow the Abstract in your revised manuscript. This text is subject to editorial change and should be distinct from the scientific abstract. Please see our author guidelines for more information: https://journals.plos.org/plosmedicine/s/revising-your-manuscript#loc-author-summary

8. In text citations: Please place reference citations within square brackets, placed before the sentence punctuation, for example [1,2]. Where multiple references are listed, please do not include spaces within brackets.

9. Introduction: “A protocol was pre-registered before having access to the data and a webpage has been designed to visualize all results (https://nbepi.com/atlas).” Please move this information to the Methods.

10. Methods: “We designed a population-based cohort study including all 7,378,598 persons living in Denmark at any point between January 1, 2000 and December 31, 2018” Are more recent data available?

11. Methods: Statistical analysis: Please clarify if comorbidity status or level of education were taken into account in the analyses.

12. Methods: Given Reviewer 1’s comment about underlying social determinants of health, and the likelihood that these are geographically clustered, please explain if clustering by region or deprivation index was taken into account.

13. Methods: Please include a copy of the pre-specified analysis plan for the study as a supporting information file.

14. Methods: Please report your data according to GATHER (or according to the most relevant guideline for your study) and enclose a completed GATHER checklist as a supplementary document. Please add the following statement, or similar, to the Methods: "This study is reported as per the Guidelines for Accurate and Transparent Health Estimates Reporting (GATHER) guideline (S1 Checklist)." Please see http://gather-statement.org/ for more information.

15. Results: Please clarify which subchapters are being described here: “When looking

within ICD chapters (Figure 5b), there is considerable heterogeneity, with MRRs ranging from 0.50 (0.38 to 0.64) to 152.31 (116.61 to 198.93); and LYLs ranging from -3.37 (-3.48 to -3.24) years to 26.95 (25.43 to 28.58) years.”

16. Discussion: Please present and organize the Discussion as follows: a short, clear summary of the article's findings; what the study adds to existing research and where and why the results may differ from previous research; strengths and limitations of the study; implications and next steps for research, clinical practice, and/or public policy; one-paragraph conclusion.

17. Figure 1 and Figure 2: If possible, please also provide these data in table format.

Comments from the reviewers:

Reviewer #1: General comments: 

The study authors have created a fantastic resource exploiting the fantastic ability in Denmark to link health service records to death. This will be a resource for many users and particularly for burden of disease estimation. I only opened the website after reading the full paper and found it a much greater resource than I could have imagined from just reading the paper. It seems that you are 'under-selling' what you are giving the world! The least I would suggest you can do is provide an example for one particular condition showing all the results that you can find in the tool after clicking on that specific disease. I have a lot of more specific comment below, largely asking for more precise detail but a few more conceptual things and limitations that you have not spun out in enough detail.

Specific comments:

Abstract: I would not call life expectancy an 'absolute-risk mortality metric'

Abstract: how did you compute the 7.4 million persons living in Denmark 2000-2018? From a statement a bit further it seems that you conceptualise is as "anyone who lived in Denmark for some part of all of 2000-2018'

Abstract: you do not define LYL

Introduction: The statement "When looking at life expectancy, the Global Burden of Disease (GBD) studies measure Years of Life Lost (YLLs),3 which estimates the potential years of life lost." suggests that YLLs are a measure of life expectancy which it is not

Page 7 line 2: what do you mean with 'admissions to ….outpatient facilities'? Do you mean e.g. day surgeries or also consultations with medical specialists?

Page 8: you estimate your MRRs and difference in average life expectancy against population all-cause mortality rates (although I am a little confused by the contradicting statement on page 10: "mortality rates after diagnosis remained higher for those with a diagnosis of any of the disorders, compared to those without the diagnosis"). These population mortality rates also include the deaths from any condition of interest. Conceptually, I think you are using a counterfactual approach: "if a person had not become a case of a disease, how different would this person's risk of death and remaining life expectancy have been?". To do that correctly you would need to contrast people with a disorder with the rest of the population without the disorder. Most disorders will be a rare enough reason for death to make such comparisons a reasonable proxy for a true RR or difference in life expectancy but you also include disorders such as IHD and stroke that are highly prevalent/incident at oldest ages and then it would not be a very good proxy. 

Results: when you present prevalence of aggregates of the 39 selected disorders, are you conceptualizing that as an individual experiencing any of the more specific disorders?

Discussion page 12 top: you mention dementia here but it is not one of the 39 chosen conditions that you concentrate on in this paper (…and see comment above: did you take the different spots in ICD where dementia is coded?)

Discussion page 12: what future study are you planning to further explore a potential impact of air pollution?

Discussion page 12: what do you mean with 'late-onset conditions'? For dementia that would read as onset at older ages but for cancer, that may be the case for some types of cancer but certainly not generalisable

Discussion page 12: the single sentence about COVID-19 seems a little gratuitous: either expand on the topic or leave it out.

Limitations: you do not mention one important limitation: a finding of excess mortality risk or reduced life span associated with a particular diagnosis may not be related to the disease per se but reflecting common underlying risks even if there is no evidence of a direct relationship. Many diseases are linked with upstream risks like poverty or poor education. These in turn cause many other more proximal risks to be more common. If the outcome of interest has a link to poverty it takes on the baggage of all other risks that are elevated with poverty leading to excess deaths that are not 'due to' the condition of interest but confounded by the excess baggage of risk factors this person carries. In other words, you may not be estimating a true counterfactual: the absence of a disease of interest may not take away the fact that someone is poor or has low education and therefore lots of other things predisposing to premature mortality.

Limitations: you mention that you only capture diagnoses from hospital encounters and that you may therefore be missing conditions for which people largely seek care with GPs or do not seek care. A consequence for those conditions is that you may be selecting more severe cases of the disease who are more likely to appear in your disease registry and hence overestimate their mortality risk.

Appendix: by choosing the whole F chapter as 'mental disorders' you partially include dementia but not cases coded to G30 and G31. It will depend on local coding practices which codes are preferred. There are more examples of disease that straddle different ICD chapters and certainly those that fall across multiple smaller ICD groupings you created.

Appendix: your list of ICD categories uses mainly the S and T chapters with 'nature of injury' codes but you add 'cause of injury' codes for suicide and violence. There would be overlap between those: e.g. someone getting assaulted (X59-Y09) with as a consequence a head injury (S00-S09): that would be one individual and one injury episode.

Reviewer #2: This study tried to estimate a wide range of health metrics in Denmark from 2000 to 2018 using a comprehensive population-based cohort including all residents of Denmark. Although some estimates produced in this study are useful for policy making and population health research, I don't think there is enough innovation and unique contribution for a original research paper. Most of the health metrics (the important ones) estimated in this study have already been estimated by IHME using a more comprehensive and rigorous approach. I believe that GBD studies also utilized this national cohort for their Danish estimates. I doubt this study make much additional contribution. In addition, I found one major assumption about remission made in this study particulary contraversial, which makes the results less reliable or useful. 

Here are my specific comments

1. Please provide line numbers for easier reference

2. Data used in this study are not publicly available and some restrictions may apply

3. Page 5, Introduction: "There is a need to better understand the impact on life expectancy of non-fatal disorders…" GBD studies also produced years of lives lost due to disability (YLDs) and thus DALYs, which can reflect burden due to the 'upstream' causes or risk factors. Actually, when calculating the YLDs, the incidence/proportion of different stages of a condition (four stages of cancers) and their coresponding disablity weights were all carefully estimated to produce proper YLDs for each condition. So, I don't think this paragraph is well grounded. 

4. Page 5, Introduction: "…have assumed a fixed age-of-onset of 15 years…" So, if age-of-onset is 15 years, does it mean the age of onset is 15 years old? It's a bit unclear. It's also really hard to believe that age of onset for mental disorders is 15 years old.

5. Page 6, Introduction: "limited range of mortality-related estimates (e.g. the GBD only presents YLLs)." That's not true. GBD studies actually produced a wide range of health metrics, including YLLs, YLDs, DALYs, all-cause and cause-specifc mortality rates, etc. 

6. Page 7 and page 13: The authors ignored remission/recovery period for all diseases (one of their assumptions). This assumption does not make much sense for most infectious diseases that does not last long, do not affect health once fully recovered and can re-occur to the same person multiple times.

7. The auhtors found that air pollution has little effects on the mortality rates. However, since Denmak does not have severe air pollution issue, I don't think this finding is true for many developing countries where air pollution is moderate or poor. 

Reviewer #3: This is a useful, innovative way to examine disease burden in high-income countries.

The MRRs are reasonably clear in methodology and presentation. However, the calculations for LYLs and the use of the method published earlier is not clear, and leaves the reader hanging. Further details of the calculations and a worked example in the appendix would be helpful. An appendix table showing the difference for say 39 conditions using the LYL used here versus a fixed LYL approach would be helpful- how does the new method alter priority setting?

As well, the main uncertainty here will not be sampling, which leads to the narrow CIs for most conditions, but two factors: (i) Misclassifications of the causes of death particularly at older ages, where COD data are less certain- so some exploration of the Danish death registry data for ill-defined contributions to COD by age would be helpful, and if possible stratifying the analyses into causes with low misclassification and those with high may be a useful appendix; (2) As mentioned, but not detailed in any sensitivity analyses, the variation in age at onset for conditions. 

Minor point that the discussion results for COPD versus vascular should be in the results, and not presented in the discussion.

Finally, Figure 5 is way too complicated , it should either be simplified- say to the top 20 leading causes of death, or leading one in each ICD10 chapter.

[LINK]

---

## [Decision Letter · Decision Letter 2]

5 May 2022

Dear Dr. Plana-Ripoll,

Thank you very much for re-submitting your manuscript "An atlas of mortality metrics associated with a comprehensive range of disorders. A Danish cohort study" (PMEDICINE-D-21-05097R2) for review by PLOS Medicine.

I have discussed the paper with my colleagues and the academic editor and it was also seen again by three reviewers. I am pleased to say that provided the remaining editorial and production issues are dealt with we are planning to accept the paper for publication in the journal.

[LINK]

We look forward to receiving the revised manuscript by May 12 2022 11:59PM.   

Sincerely,

Caitlin Moyer, Ph.D.

Associate Editor 

PLOS Medicine

plosmedicine.org

Requests from Editors:

1. Title: Please revise the title to: “Analysis of mortality metrics associated with a comprehensive range of disorders in Denmark, 2000-2018: A population-based cohort study” or similar. Please update the title in the text as well as the manuscript submission system.

2. Abstract: Line 98: Please refer to an association between air pollution and MRRs, rather than an effect of air pollution.

3. Abstract: Methods and Findings: Line 107: Please make the sentence describing the limitations of the study more obvious. We suggest: “A limitation of the study is the possibility that the association between different disorders and mortality could be explained by underlying factors associated with both the disorder and mortality.” We suggest moving the statement about the adjustment for air pollution to the end of the results description.

4. Author summary: Please re-format as bulleted points (3-4 per section) rather than in paragraph format. We suggest:

Why Was This Study Done?

-There have been many studies related to mortality linked to particular disorders, but these studies have not covered a comprehensive range of disorders.

-Previous studies have traditionally focused on relative measures of mortality (e.g. mortality rate ratios) or crude estimates of life expectancy that do not incorporate variation in the age of onset of the disorder.

-Here the researchers address these issues in a comprehensive atlas of mortality estimates based on Danish registers.

What Did the Researchers Do and Find?

-Based on 7,378,598 persons living in Denmark in 2000-2018, the researchers used national registers to identify individuals diagnosed with 1,803 specific categories of disorders.

-For each of these disorders, a panel of epidemiological and mortality metrics was estimated, including incidence rates, age-of-onset distributions, mortality rate ratios (MRRs) and life years lost (LYLs).

-Within a set of 39 selected medical conditions, mortality rates were larger and life expectancy shorter for 37 conditions compared to the Danish general population.

-The researchers have prepared an interactive data visualization to optimize the interrogation of their findings (http://nbepi.com/atlas).

What Do These Findings Mean?

-This study allows a more fine-grained analysis of the associations between a comprehensive set of disorders and mortality-related estimates.

-These findings can guide health research and serve as a benchmark to evaluate future health interventions.

5. Introduction: Line 183-184: We suggest “Furthermore, differences between study designs complicates direct comparison of mortality between different disorders.” if accurate.

6. Methods: Analysis plan and Protocol: Line 195 and 276: Thank you for making the protocol and analysis plan publicly available. Please mention the included protocol and analysis plan in the supporting data files consistently throughout the text (e.g. S2_Text). Rather than including the web link in the text, we suggest including a reference for https://doi.org/10.17605/OSF.IO/ZAFHU.

7. Results: Line 345-347: Rather than including the web links for the data and interactive webpage in the text, we suggest including these in the reference list.

8. Results: Line 330-331: “Sex-specific estimates are available in supplementary S4 Figure.” If possible, please briefly summarize the key findings from the sex-specific analyses.

9. Author Contributions: Line 451: Please remove the “Authors’ Contributions” section from the main text and be sure all information is entered completely and accurately into the manuscript submission system.

10. References: Please check the formatting of each reference in the list. Please use the "Vancouver" style for reference formatting, and see our website for other reference guidelines https://journals.plos.org/plosmedicine/s/submission-guidelines#loc-references

11. Journal name abbreviations should be those found in the National Center for Biotechnology Information (NCBI) databases. For example, please use “Lancet” as the journal title for references 1 and 3. Please use Br J Psychiatry as the journal title for reference 2. For reference 10, “PLoS One” should be the title. Please check the formatting throughout.

12. Figures 2, 3, 4, 5 and Figure S2: In the legends, please note the meaning of the red dashed line.

13. Table S2: Thank you for including the GATHER checklist. Please do not refer to the page numbers, and instead please replace these with paragraph numbers per section (e.g. "Methods, paragraph 1").

14. Figures S2, S3, S4: If possible, please also present these data in table format (similar to the presentation of Table S4).

Comments from Reviewers:

Reviewer #1: I think the authors have done a good job at responding to comments.

A few minor remaining issues:

Reviewer 1, comment 3: spelling out an acronym is not same as defining

Reviewer 1, comment 6: I'm largely OK with explanation but would expect this to be discussed in limitation section of discussion rather than hidden somewhere in appendix text.

Reviewer 1, comment 8/14: you have not really addressed the issue of dementia codes that straddle two ICD chapters

Reviewer 2, comment about GBD: agree with most of the response. The statements on remission look a little odd; we have not used Dismod 2 for 15 years and I don't understand the reference to remission being a 'top-down process'

Reviewer #2: I think the authors have adequately answered the reviewers' questions and addressed our concerns. I am satisfied with the authors' responses and happy to accept this revised version. Congratulations!

Reviewer #3: The authors have addressed most of my concerns. I quibble about the still too complicated Figure 5, but defer to the authors and editors to make this decision.

[LINK]

---

## [Editor Report · Decision Letter 3]

17 May 2022

Dear Dr Plana-Ripoll, 

On behalf of my colleagues and the Academic Editor, Theo Vos, I am pleased to inform you that we have agreed to publish your manuscript "Analysis of mortality metrics associated with a comprehensive range of disorders in Denmark, 2000-2018: A population-based cohort study" (PMEDICINE-D-21-05097R3) in PLOS Medicine.

Please also address the following editorial requests:

-Abstract Line 97: Please revise slightly to: “...differences in life expectancy between those with the specific condition and those in the general population (estimated life years lost (LYLs).” 

-Abstract: Line 99: Please revise to: “...to explore potential associations with MRRs.”

-Introduction: Line 185-186: Please change “complicates” to “complicate” in this sentence.

-Methods Line 198 and Line 283: Please reference the webpage, rather than providing the link in the text.

-Methods: Line 220: In response to reviewer 1, comment 8/14, please add a sentence to the Methods (around line 220) that there do exist health conditions defined from combinations of ICD-10 classification, such as dementia. Because dementia is used as an example, please mention in the text the definition for dementia (e.g.“Dementia was included as different specific 3-level ICD-10 codes [include the codes here]”).

-Methods: Line 257: In response to reviewer 1, comment 6, please add the explanation from the Supporting Information Text here describing the use of the general population rather than those without the disease as the comparison group, as this is helpful to understanding the LYL metric: “The main reason to compare those with a given disease to the general population – and not to persons without the disease– is that the number of Life Years Lost at a given age, e.g. 45 years, is estimated using mortality rates at ages 45 years and beyond. By choosing persons without the disease as a comparison group, we would assume that someone who has not experienced the disease at age 45, would remain free of the disease until death. Although it might seem problematic to include persons with a disease in both the diseased and reference groups, this is analogous to widely used (and classic) standardized mortality ratios (SMRs), which compare mortality in a group of persons to the one in the general population. In any case, differences in life expectancy would be even larger if the comparison group were persons without the disease.”

Reference 5: Please change the journal title to PLoS One.

PRESS

Sincerely, 

Caitlin Moyer, Ph.D. 

Associate Editor 

PLOS Medicine